# Molecular Dynamics Simulations of the Proteins Regulating Synaptic Vesicle Fusion

**DOI:** 10.3390/membranes13030307

**Published:** 2023-03-06

**Authors:** Maria Bykhovskaia

**Affiliations:** Neurology Department, Wayne State University, Detroit, MI 48202, USA; mbykhovs@med.wayne.edu

**Keywords:** synaptotagmin, SNARE complex, complexin, lipid bilayers, exocytosis, neuronal transmitters

## Abstract

Neuronal transmitters are packaged in synaptic vesicles (SVs) and released by the fusion of SVs with the presynaptic membrane (PM). An inflow of Ca^2+^ into the nerve terminal triggers fusion, and the SV-associated protein Synaptotagmin 1 (Syt1) serves as a Ca^2+^ sensor. In preparation for fusion, SVs become attached to the PM by the SNARE protein complex, a coiled-coil bundle that exerts the force overcoming SV-PM repulsion. A cytosolic protein Complexin (Cpx) attaches to the SNARE complex and differentially regulates the evoked and spontaneous release components. It is still debated how the dynamic interactions of Syt1, SNARE proteins and Cpx lead to fusion. This problem is confounded by heterogeneity in the conformational states of the prefusion protein–lipid complex and by the lack of tools to experimentally monitor the rapid conformational transitions of the complex, which occur at a sub-millisecond scale. However, these complications can be overcome employing molecular dynamics (MDs), a computational approach that enables simulating interactions and conformational transitions of proteins and lipids. This review discusses the use of molecular dynamics for the investigation of the pre-fusion protein–lipid complex. We discuss the dynamics of the SNARE complex between lipid bilayers, as well as the interactions of Syt1 with lipids and SNARE proteins, and Cpx regulating the assembly of the SNARE complex.

## 1. Introduction

Neurons communicate by releasing neuronal transmitters into the synaptic gap. Transmitters are packed in synaptic vesicles (SVs) and released by the fusion of SVs with the presynaptic membrane (PM). The attachment of an SV to the PM is mediated by the SNARE complex [1,2,3], a coil-coiled four-helical bundle, which consists of the SV protein synaptobrevin (Sb) and the PM-associated proteins syntaxin 1A (Sx) and SNAP25, or t-SNARE. The assembly of the SNARE bundle enables overcoming the electrostatic and hydration repulsion between the SV and PM lipid bilayers [4].

Rapid synchronous fusion of SVs with PM is triggered by an influx of Ca^2+^ ions into the nerve terminal. An SV-associated protein Synaptotagmin 1 (Syt1) acts as a Ca^2+^ sensor, and evoked synaptic transmission is completely abolished in the absence of Syt1 [5,6]. Syt1 comprises two Ca^2+^ binding domains, C2A and C2B, which are attached to an SV by a transmembrane helix [7]. Each domain has two loops forming a Ca^2+^ binding pocket, and in each of the pockets, Ca^2+^ ions are chelated by five aspartic acids [8,9]. It is agreed that synergistic coordinated insertion of the tips of the C2 domains into the phospholipid membrane drives fusion [5,10,11], but other mechanistic details of Syt1 action are still debated.

Syt1 interacts with the SNARE complex, and multiple studies suggest an important role for Syt1–SNARE interactions during fusion [12,13,14,15,16,17]. However, other studies have argued against this possibility [18,19], and it is still debated how the SNARE–Syt1 complex is formed in vivo and what the role of Syt1–SNARE interactions is in the fusion process.

The fusion is tightly regulated by the cytosolic protein Complexin (Cpx), which attaches to the SNARE bundle [20] and serves as a positive regulator of synchronous release, promoting and accelerating evoked synaptic transmission [21,22,23,24,25,26]. The effect of Cpx on synchronous fusion is Ca^2+^-dependent [21] and several studies suggested a functional [26,27,28,29,30] or molecular [31] interaction between Cpx and Syt1. Cpx deletion also produces a drastic increase in spontaneous Ca^2+^-independent transmission [24], suggesting that the energetic barrier for SV fusion is reduced in the absence of Cpx [32,33]. It has been established that different domains of Cpx control evoked spontaneous transmission and that these two Cpx functions are decoupled [34,35]. The inhibitory role of Cpx in spontaneous transmission was extensively studied in vitro [36,37,38,39] and in vivo [24,34,40], and several competing models of the Cpx clamping function have been developed. However, it remains obscure how Cpx promotes and synchronizes the evoked transmission.

The proteins regulating synaptic fusion have been extensively studied with tools and perspectives of biochemistry and molecular biology, and tremendous progress has been achieved in understanding their interactions [2,18,41,42,43,44,45]. However, the atomistic details of the dynamic Syt1-SNARE-Cpx interactions are still debated, and a systematic approach to manipulating the fusion machinery and understanding disease-relevant mutations is still missing. One complication to this problem is that fusion occurs at a sub-millisecond timescale, and the underlying conformational transitions of the pre-fusion protein–lipid complex occur much faster, probably at a timescale of microseconds or tens of microseconds. Currently, such rapid conformational transitions cannot be monitored experimentally. However, they can be observed in silico employing molecular dynamics (MDs) simulations. In the present review, we discuss how MD simulations of proteins and protein–lipid complexes promoted our understanding of the protein dynamics regulating SV fusion.

## 2. The SNARE Complex Assembly

The fully assembled SNARE complex is a multicomponent molecular system, which encompasses a four-helical coil-coiled bundle, transmembrane (TM) domains of Sb and Sx, the palmitoylated loop of SNAP25, and the N-terminal domain of Sx, which attaches to PM [46]. Since zippering of the four-helical bundle is thought to provide the force to counterbalance the SV-PM repulsion, the mechanics and dynamics of the SNARE bundle assembly have been studied extensively. Crystallography studies [47] demonstrated that the bundle has distinct layers (Figure 1A), and the initial all-atom MD study (AAMD) [48] showed that a compact and stiff bundle has limited conformational dynamics. The latter study also showed that the bundle is largely stabilized by electrostatic forces, although the hydrophobic interactions add to the bundle rigidity.

Although the AAMD method was instrumental for the initial investigation of the dynamics of the bundle [48,49], as well as of the membrane insertion of the TM and linker domains of the SNARE proteins [50,51], the size of the molecular systems and timescales handled by the AAMD approach remained a limitation. Therefore, coarse-grain MD approaches (CGMD) were developed to simulate zippering of the SNARE complex between lipid bilayers. The initial CGMD simulations of the SNARE complex interacting with lipids [52,53,54,55] have been performed employing MARTINI force field [56]. This approach was employed to model fusion mediated by four SNARE complexes, starting from all the SNARE bundles being in a nearly assembled state (up Layer 5). These CGMD simulations enabled observation, in silico, of the final stages of SNARE zippering that trigger fusion, including lipid stalk formation and pore opening (Figure 1B) [57].

To investigate the assembly of the entire SNARE bundle and to understand how it depends on the number of the SNARE complexes attaching an SV to the PM, customized CGMD force fields were developed [58,59,60,61]. Indeed, it has been shown that the CGMD force fields, including MARTINI, are not suited to a wide range of applications, and they need to be refined and customized for specific molecular systems [62,63,64,65,66].

The customized CGMD approaches [58,59,60,61] modeled SNARE proteins as sequences of beads, each bead representing either a single amino acid [60] or a chain of four amino acids [61]. These studies did not model lipid bilayers explicitly but instead represented the PM and SV membranes as a continuum excreting electrostatic, hydration repulsion, and mechanical tension forces. Both models demonstrated, in silico, that SNARE zippering counterbalances membrane repulsion, and that increasing the number of SNARE complexes from one to three significantly accelerates fusion. Strikingly, both models also revealed that a further increase in the number of the SNARE complexes adds very little to the adhesive forces bringing together an SV and the PM [60,61] (Figure 1C). These findings were in agreement with experimental studies, which suggested that several SNARE complexes are likely to mediate synaptic fusion [67,68,69], even though under certain experimental conditions, a single SNARE complex may be sufficient [70].

Interestingly, the simulations of the kinetics of the SNARE zippering at various initial separations between the C-termini of Sb and Sx showed an exponential relationship between the number of the initially unraveled helical turns and the assembly times [59]. Notably, it was shown that the assembly of two or three membrane-proximal layers in the SNARE complex would take tens of nanoseconds, while the assembly of the entire bundle could take microseconds (Figure 1D).

Together, these findings suggest that three to four SNARE complexes in a nearly assembled state, with only several membrane-proximal layers being separated, would represent the most efficient prefusion complex, which could fully assemble at a sub-microsecond timescale.

## 3. Cpx as a Dynamic Fusion Clamp

The SV-PM fusion and release of transmitters can occur spontaneously, independently of Ca^2+^ influx. The spontaneous fusion can be clamped by Cpx [24,25,45], and it can be drastically promoted in Cpx-deleted synapses, which is especially prominent in invertebrates. Numerous studies suggested that the interaction of Cpx with the SNARE proteins inhibits the SNARE assembly [32,33,34,36,39,71,72]; however, the atomistic detail of this mechanism is still debated.

Cpx includes the central and accessory alpha helixes, as well as the C-terminal and N-terminal domains, which are largely unstructured, and the crystallography studies [20] demonstrated that Cpx binds the SNARE bundle via its central and accessory helixes (Figure 2A). Notably, it was also shown that the accessory helix predominantly contributes to the clamping mechanism [35]. Several competing models for the Cpx clamping function have been proposed, which implied that Cpx accessory helix either competes with Sb for the SNARE binding [33,36,39,71] or destabilizes the Cpx central helix [40].

To investigate, in silico, the role of Cpx in the SNARE assembly, the AAMD simulations of Cpx interacting with the partially unraveled SNARE bundle were performed [72]. Interestingly, this study revealed that the Cpx accessory helix could interact with the unstructured C-terminus of Sb, preventing it from zippering onto the core t-SNARE bundle, thus stabilizing the partially assembled structure of the SNARE complex with two or three of its C-terminal layers being unraveled (Figure 2B).

Subsequently, this model was extended to incorporate the SNARE-Cpx interactions with lipid bilayers mimicking the PM and an SV [73]. These AAMD simulations showed that the Cpx accessory helix could also act as a barrier between the SV and the SNARE bundle, thus hindering PM-SV fusion (Figure 2C). Importantly, this model enabled making several valid predictions for the poor-clamp and super-clamp mutations in Cpx and Sb [73,74].

Together, the AAMD simulations outlined above and coupled with in vivo studies [72,73,74] suggested that the Cpx accessory helix may simply act as a spacer between an SV and the SNARE bundle, in addition stabilizing the unstructured C-terminus of Sb, thus preventing spontaneous full SNARE assembly and PM-SV fusion.

**Figure 2 membranes-13-00307-f002:**
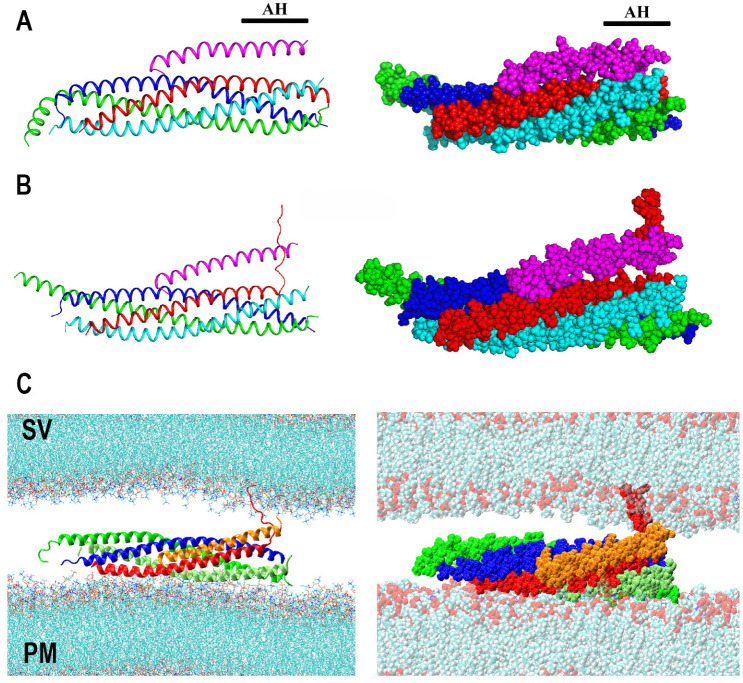
The model of Cpx clamping function driven by AAMD simulations. (**A**) Two representations of the SNARE-Cpx complex. The structure was obtained by crystallography and equilibrated by AAMD. Reproduced from [72]. Blue: Sx; red: Sb; green: SN1; cyan SN2; magenta: Cpx. AH: Accessory helix. (**B**) Cpx stabilizes a partially unraveled state of Sb (Layers 6–8). Reproduced from [75] with permission (license 5472810090395). (**C**) The partially unraveled SNARE complex between lipid bilayers mimicking an SV and the PM. Note that Cpx (orange) creates a barrier between the SNARE bundle and the SV via its accessory helix, in addition to stabilizing the partially unraveled state of Sb.

## 4. Syt1 and Its Interaction with Lipid Bilayers

Syt1 triggers fusion upon Ca^2+^ binding, presumably by inserting the Ca^2+^-bound tips of its C2 domains into the lipid bilayer(s) [41]. However, the atomistic mechanics and dynamics of this process are still debated. The crystallography study showed that the C2A and C2B domains of Syt1 are tightly coupled and perpendicularly oriented [76]. However, optical studies suggested that in the solution, Syt1 may sample multiple conformations and the interactions with lipids would likely affect the Syt1 conformational ensemble [77,78,79].

The conformational space of Syt1 has been investigated employing AAMD coupled with Monte Carlo sampling of the C2 domain orientations [80]. This study identified several conformational states of the Syt1 C2AB tandem, all having tightly coupled C2 domains. Notably, this study also showed that Ca^2+^ binding decouples the C2 domains and allows them to rotate more freely, accelerating Syt1 conformational transitions.

Since the immersion into lipid bilayers is thought to be the major mechanism driving fusion [41], several studies employed AAMD to investigate the interactions of Syt1 C2 domains with lipids [81,82,83]. Initially, it was shown that binding the C2B module to the lipid bilayer drives lipid bending [81]. This finding supported the hypothesis that Syt1 drives formation of the stalk between lipid bilayers by promoting membrane curvature [11]. Subsequent studies [82,83] modeled the interactions of Syt1 domains with the PM bilayer by incorporating anionic lipids and phosphatidylinositol 4,5-biphosphate (PIP2), which is an essential component of the PM. These studies demonstrated that the C2B domain forms strong attachments to the PM via its Ca^2+^ binding loops and the polybasic motif (Figure 3A), in agreement with molecular biology and spectroscopy experiments [84,85,86,87]. Notably, both studies [82,83] demonstrated that the C2B domain does not robustly associate with the SV bilayer lacking PIP2, while the C2A domain does bind the SV bilayer via its Ca^2+^ binding loops (Figure 3B). These studies also demonstrated that the C2A domain robustly binds the PM bilayer (Figure 3C).

The AAMD studies cited above were consistent with two possibilities for Syt1 dynamics upon Ca^2+^ binding: (1) the C2AB tandem bridges the PM and an SV [78,88] and (2) both domains immerse into the PM, thus promoting PM curvature [11,89,90]. To discriminate between these possibilities, prolonged AAMD simulations of the C2AB tandem between lipid bilayers mimicking an SV and the PM were then performed at a microsecond scale [83]. This study demonstrated a conformational transition of the Syt1 C2AB tandem from the PM-SV bridging to the PM-attached conformation, suggesting that the second scenario is more likely.

Interestingly, a latter study [83] also showed that the C2 domains do not cooperate in penetrating into PM but rather preclude each other from deep immersion into lipids. Indeed, the isolated C2 domains immersed into the PM deeper than when being attached within the C2AB tandem (Figure 3D). These findings suggested that the C2 domains of Syt1 need to be decoupled within the prefusion protein complex, driving the hypothesis that the interactions with other components of the protein fusion machinery serve to uncouple the C2 domains of Syt1. One possibility is that the interactions of Syt1 with the SNARE bundle carry out this function.

## 5. The Prefusion Syt1-SNARE-Cpx Complex

Spin labeling studies demonstrated that the Syt1-SNARE complex samples multiple conformational states in the solution [91]. Consistently, multiple interfaces between the C2B domain and the SNARE complex were revealed by crystallography, including an extensive primary conserved interface [15]. Interestingly, a different C2B-SNARE interface was identified by the NMR approach [17]. These findings warranted systematic in silico studies of the Syt1-SNARE complex.

To sample the conformational space of the Syt1-SNARE complex, prolonged AAMD simulations were performed and coupled with in silico docking [83]. This study identified three different conformational states of the Syt1-SNARE-Cpx complex (Figure 4A), which were stable at a microsecond scale. The C2B-SNARE interface of the State 3 matched the primary conserved interface discovered by crystallography [15]. Interestingly, two of the three states had Syt1 directly interacting with Cpx (Figure 4A, States 1 and 2). The latter finding was in line with multiple experimental studies, which suggested a functional [26,27,28,29,30] or molecular [31] interaction between Cpx and Syt1 in vivo.

How does the association with the SNARE bundle affect the ability of the C2 domains of Syt1 to penetrate into lipid bilayers? The AAMD simulations [83] revealed that when the Syt1-SNARE-Cpx complex (State 3), with Syt1 being in its Ca^2+^-bound state, interacted with the lipid bilayer mimicking PM (Figure 4B), the tips of both C2A and C2B domains penetrated into the PM significantly deeper than within the isolated C2AB tandem interacting with the PM (Figure 4C). In other words, the attachment of the C2B domain to the SNARE bundle uncoupled the C2A and C2B domains and, consequently, promoted the insertion of the tips of both C2 domains into the lipid bilayer mimicking the PM.

How does the Ca^2+^Syt1-SNARE-Cpx pre-fusion complex trigger SV-PM fusion? To elucidate this question, AAMD simulations of the prefusion protein complex between lipid bilayers mimicking the PM and an SV were performed [83]. This study demonstrated, in silico, that this complex in its Ca^2+^-bound form enables the insertion of the tips of C2 domain of Syt1 into the PM, thus promoting PM curvature and also firmly anchoring the t-SNARE bundle to the PM, acting synergistically with SNARE zippering and driving the SV-PM merging (Figure 4D,E). In contrast, the same molecular system in the absence of Ca^2+^ did not promote fusion (Figure 4E). In summary, this study [83] identified the conformation of the minimal protein machinery (Figure 4B,D) capable of driving SV-PM fusion.

Subsequently, AAMD simulations were performed for the system containing several Syt1-SNARE-Cpx complexed between the PM and SV lipid bilayers (Figure 5A) [92]. Importantly, this study showed that even in the case of multiple SNARE complexes, the C2B domains of Syt1 robustly bind the SNARE bundles via their primary interfaces identified by crystallography [15]. In contrast, the interactions of the C2A domain showed some heterogeneity: in the end of the trajectory, two Syt1-SNARE-Cpx complexes had the C2A domains penetrating into the PM, while the other two complexes had the C2A domains bridging to the SV and interacting with Cpx (Figure 5B). This finding can be interpreted either as heterogeneity in Syt1-SNARE-Cpx conformational states within the prefusion protein–lipid complex or, alternatively, as dynamic intermediate states corresponding to the conformational transitions of the complexes to their final prefusion states. More prolonged AAMD simulations will be needed to discriminate between these possibilities.

## 6. Conclusions and Further Directions

MD simulations of the SNARE proteins, Syt1, and Cpx elucidated the mechanistic detail of the final stages of SNARE zippering, enabled the development of the all-atom model of the fusion clamp, and revealed the atomistic detail of Syt1 immersing into lipid bilayers and triggering fusion (Table 1). As the developments in supercomputing enable more prolonged AAMD simulations of larger molecular systems [93,94,95,96], the dynamics of synaptic fusion will be further elucidated.

Notably, in silico studies, in particular prolonged AAMD simulations at a microsecond scale, can elucidate how the fusion proteins transition to their pre-fusion states. Both in vitro [91] and in silico [92] studies suggest that the Syt1-SNARE complex is heterogeneous, which may reflect the dynamic conformational transitions of the pre-fusion Syt1-SNARE-Cpx complex to its final state triggering fusion, which occur in vivo. The timescale of such transitions would likely occur at a scale of microseconds or tens of microseconds, and, therefore, cannot be monitored experimentally. However, the dynamics of such conformational transitions can be captured in silico.

Importantly, the interactions of Syt1 and Cpx observed in silico [83] and in vitro [31] and also suggested by in vivo [26,27,28,29,30] studies may play a pivotal role in guiding the pre-fusion complex through the conformational transitions leading to fusion. Indeed, Cpx was shown to synchronize evoked release [22,24,26] by a mechanism which is distinct from clamping spontaneous fusion [34,35]. It is a plausible hypothesis that Cpx may synchronize fusion by accelerating the conformational transitions of the Syt1-SNARE complex, and in silico studies, such as AAMG or GCMD, can test this hypothesis directly.

Furthermore, the development of supercomputing capabilities makes it plausible to incorporate additional components of the protein fusion machinery and to develop the atomistic model of the pre-fusion protein dynamics beyond the minimal Syt-SNARE-Cpx complex. In particular, the Munc family of proteins was shown to orchestrate the assembly of the SNARE complex [1,2,43], with Minc18 possibly serving as a template, forming a tripartite complex with t-SNARE and Sb and stabilizing the half-zippered state of the SNARE bundle [97,98]. The in silico methods, such as AAMD or CGMD, could capture the dynamics of this process in the atomistic detail.

Finally, the CG [57,58,60] and AA [92] models of several SNARE bundles mediating fusion have set the stage for in silico studies of the SNARE self-organization, including the interactions and possible cooperation between multiple SNARE complexes. Indeed, competitive models for the interactions between SNARE bundles mediating fusion have been proposed [33,36,99], and the AAMD and GCMD methods could test the feasibility of these models in silico at the level of mechanics and dynamics of atomic interactions.

The mechanistic details outlined above could be the key for understanding numerous disease-relevant mutations in the fusion proteins, and they can be unraveled by further AAMD and GCMD studies.

## Figures and Tables

**Figure 1 membranes-13-00307-f001:**
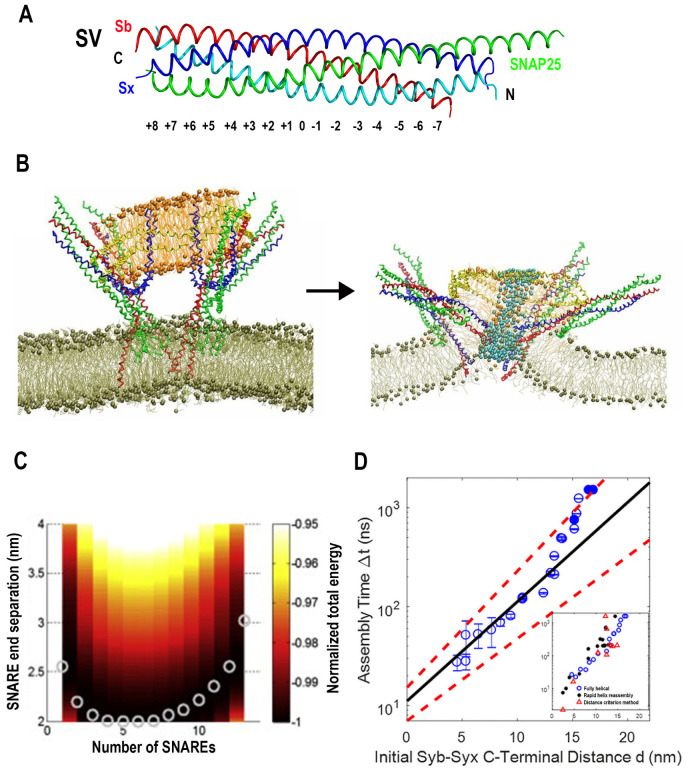
SNARE zippering. (**A**) The structure of the SNARE bundle with the denoted layers [47]. (**B**) The initial (left) and the final (right) states of the molecular system mimicking the SV and PM bilayers attached to each other by four SNARE bundles. Red: Sx, blue: Sb, green: SNAP25. Blue spheres denote water molecules diffusing through the open pore in the final state. Reproduced with permission from [57]. (**C**) The separation of an SV and the PM at equilibrium plotted against the number of the SNARE complexes mediating the SV-PM attachment [60]. Note a steep drop as the number of the SNARE complexes increases from one to two, and a further reduction in the SV-PM separation as the number of the SNARE complexes increases to three. Note also the plateau, as the number of the complexes increases further. (**D**) The assembly time of the SNARE complex depends exponentially on the initial separation of the Sb and Sx C-terminals [59]. The inset shows the results obtained using three different models of the helix assembly, which largely converge.

**Figure 3 membranes-13-00307-f003:**
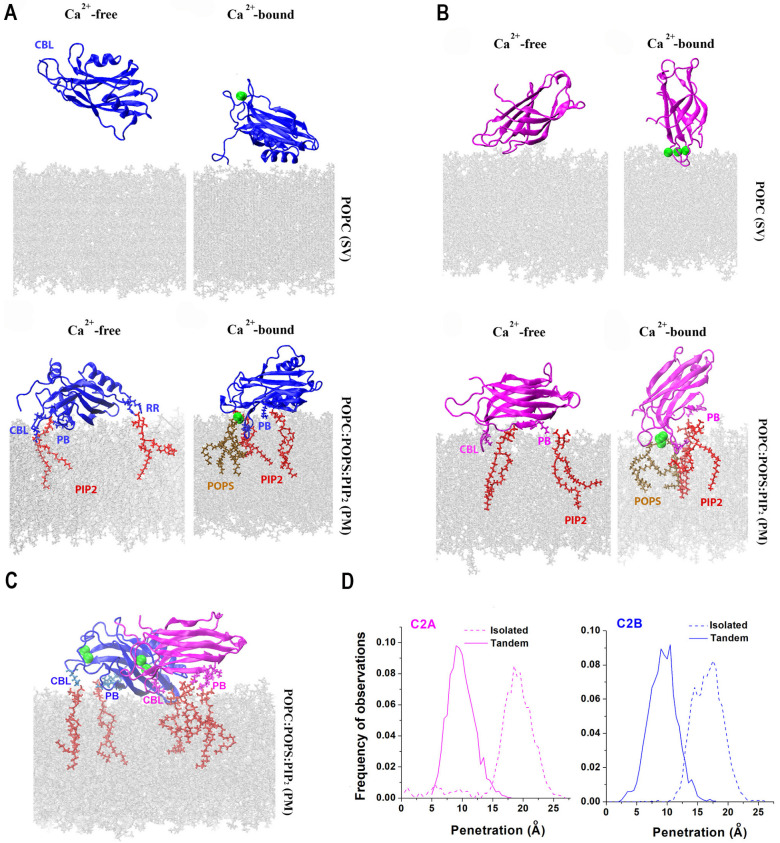
Lipid binding of the isolated domains, as well as the C2AB tandem of Syt1. (**A**) The C2B domain binds the bilayer mimicking the PM via its Ca^2+^ binding loops (CBL), polybasic stretch (PB), and the RR (Arg398-Arg399) motif opposing the CBL. Green spheres denote Ca^2+^ ions. Red: PIP2. (**B**) CBL of the C2A domain attach to either the SV or PM bilayer; however, the interaction with the PM bilayer is more extensive and the penetration into the PM is deeper. (**C**) Both C2 domains within Ca^2+^C2AB tandem attach to the PM via their CBL and PB motifs. (**D**) The penetration into the PM bilayer is deeper for the isolated Ca^2+^-bound C2 domains compared to the Ca^2+^C2AB tandem. Reproduced from [83].

**Figure 4 membranes-13-00307-f004:**
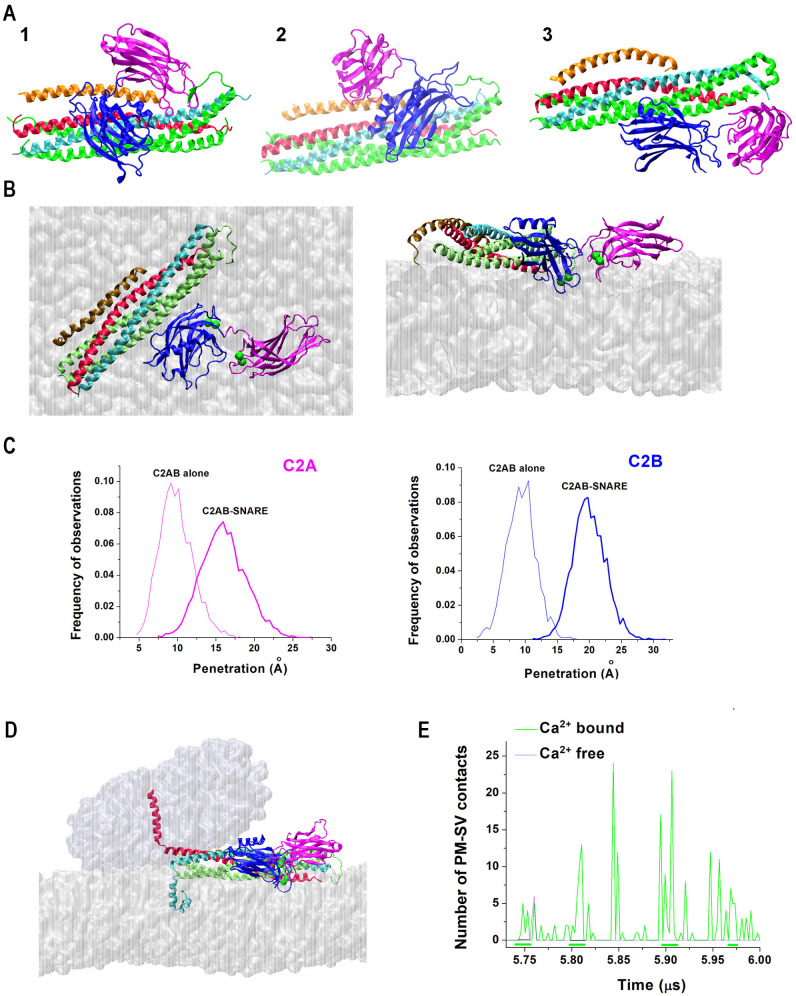
The prefusion Syt1-SNARE-Cpx complex. (**A**) Three conformational states of the Syt1-SNARE-Cpx complex obtained by AAMD simulations. Note that States 1 and 2 have Syt1 directly interacting with Cpx. (**B**) Two views of the prefusion Ca^2+^Syt1-SNARE-Cpx complex attached to the PM. Note the Ca^2+^-bound tips of C2 domains immersed into the PM. (**C**) The attachment of the C2B domain to the SNARE bundle decouples C2 domains and enables their deeper penetration into the PM. The graphs show the distributions of the penetration depths over respective 5 μs trajectories. (**D**) The prefusion Ca^2+^Syt1-SNARE-Cpx complex drives the merging of the SV (top) and the PM (bottom) bilayers. (**E**) The number of SV-PM Van der Waals contacts for the Ca^2+^-bound and Ca^2+^-free prefusion Syt1-SNARE-Cpx complexes along respective trajectories. Note continuous stretches of the SV-PM attachment for the Ca^2+^-bound complex (green lines). In contrast, for the Ca^2+^-free complex, the PM and SV bilayers are not in contact for most of the trajectory. Reproduced from [83].

**Figure 5 membranes-13-00307-f005:**
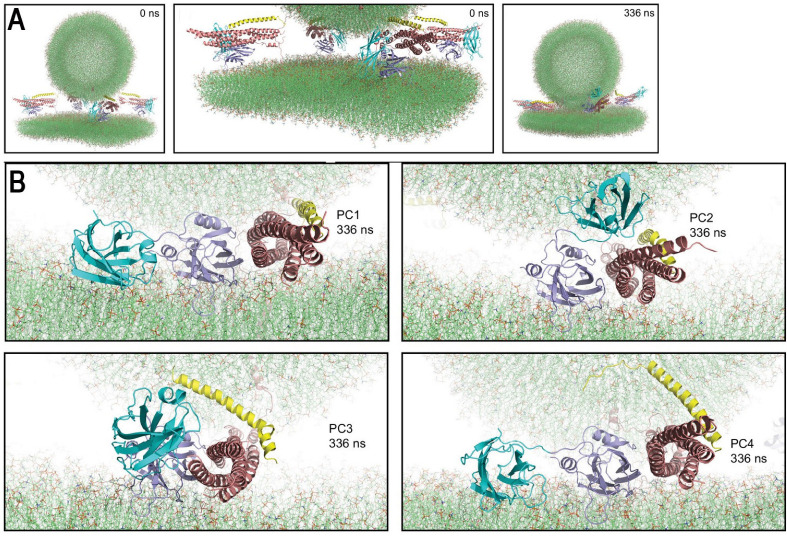
SV-PM fusion mediated by four Syt1-SNARE-Cpx complexes. (**A**) The system at the initial (0 ns) and final (336 ns) points of the trajectory. (**B**) Each of the four complexes between the bilayers of the SV and PM in the end of the trajectory. Note that all the complexes have the C2B domains (navy) attached to the SNARE bundles. In contrast, the positions of the C2A domains (cyan) vary: two complexes (PC1 and PC4) have the C2A domain attached to the the PM, while the other two complexes (PC2 and PC3) have the C2A domains interacting with Cpx (yellow) and bridging to the SV. Reproduced from [92].

**Table 1 membranes-13-00307-t001:** A summary of the major MD studies of the synaptic fusion proteins.

Main Focus	Methodology	References
SNARE bundle dynamics	AAMD	[48,49,52]
SNARE TM domains embedded in lipids	AAMD	[50,51]
	CGMD, Martini force field	[53]
SNARE zippering under the forces exerted by the PM-SV repulsion	CGMD, Martini force field	[52,57]
	CGMD, customized force fields	[58,59,60,61]
Cpx fusion clamp	AAMD	[72,73,74]
Syt1 interdomain rotations	AAMD	[16,80]
Syt1 interactions with lipids	AAMD	[81,82,83]
Syt1-SNARE-Cpx prefusion complex between PM and SV	AAMD	[83,92]

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
