# Peer review of "Molecular Dynamics Simulations of the Proteins Regulating Synaptic Vesicle Fusion"

_membranes, 2023, doi:10.3390/membranes13030307_

Round 1

Reviewer 1 Report

The article "Molecular Dynamics Simulations of the Proteins Regulating Synaptic Vesicle Fusion" presents a review on the state of synaptic vesicle fusion, especially from the perspective of simulation, using molecular dynamics. The fusion process involves different proteins, among which Synaptotagmin 1, Complexin and the SNARE complex stand out. Molecular dynamics allows obtaining information at the atomic level of the interactions that allow and give way to the fusion process, confirming in many cases information of an experimental origin.

Dr. Maria Bykhovskaia is an expert in molecular dynamics, as her contributions to the problem of synaptic vesicle fusion are relevant.

The article is well written and organized, and includes a significant number of recent references that contribute to knowledge in this area.

In general, I consider the article "Molecular Dynamics Simulations of the Proteins Regulating Synaptic Vesicle Fusion" to be an important contribution to the field and therefore recommend its publication in its current form in Membranes.

Author Response

I thank the Reviewer for the positive evaluation of the manuscript.

Reviewer 2 Report

In this paper the author reports an interesting state-of-the-art review on the computational research, especially via Molecular Dynamics (MD) simulations, that has dealt with those protein complexes associated with the fusion of synaptic vesicles, i.e., Syt1, the SNARE complex, and Cpx. Initially, attention is given to the research works focusing on how these proteins interact individually with the synaptic vesicle membrane and the presynaptic membrane, and then to the latest works where the full Syt1-SNARE-Cpx complex interacting with both membranes is investigated. Based on the review of the literature, the author tries to explain the fundamental mechanisms which might govern the process of synaptic vesicle function at the molecular level, while pointing out current knowledge gaps and limitations to overcome in future studies.

The manuscript is interesting and well written. It fits well with the aim of the “Membranes” Journal and in my opinion, it is suitable for publication almost as it is.

Just a few very minor remarks:

(1) It would be helpful to add a table in which you briefly summarize the main details and results of the most relevant works cited in the reference list. E.g., focus of the analysis: work dealing with Syt1, Cpx, SNARE, or both, etc.; methodology: MD using CG Martini, all-atom AMBER force-field, etc.; results: C2AB tandem of Syt1 has a lower penetration into the presynaptic membrane compared with the isolated C2A and C2B domains, etc.

(2) Reference [94] mentioned in the Conclusions is missing from the Reference List.

Author Response

Thank you for the positive evaluation of the manuscript. All the Reviewer's suggestions have been incorporated.

1) As suggested, I included the new table (Table 1, Conclusion, Line 277), which summarizes the major MD studies, including their focus, methodology, and references

2) The bibliography has been updated, and the reference list is now complete.